# Mapping in situ the assembly and dynamics in aqueous supramolecular polymers

Huachuan Du [1,2,3], Ruomeng Qiu[1,4], Xianwen Lou[2], Stef A. H. Jansen [2,3], Hiroaki Sai [1], Yuyang Wang[2,5], Albert J. Markvoort [2,6], E. W. Meijer [2,3,7] & Samuel I. Stupp [1,4,8,9,10] ✉

Supramolecular polymers, bonded through directional non-covalent interactions, closely mimic dynamic behaviors of biological nanofibers. However, the complexity of assembly pathways makes it highly challenging to unravel the nature of supramolecular dynamics in aqueous environments. Here we introduce a precise combinatorial titration methodology to probe in situ the assembly of peptide amphiphiles (PAs). This approach reveals a binary assembly mechanism governed by equilibrium between spheroidal micelles and β-sheet polymers. Weakening hydrogen bonding shifts the equilibrium towards micelles and decreases the internal structural order of filamentous polymers, promoting supramolecular dynamics. Extending this methodology to two-component copolymerization systems, we find a surprising tendency to form blocky nanostructures with reduced internal phase separation as the mismatch in peptide sequence decreases. Interestingly, while well-mixed copolymers acquire different dynamics, mismatched ones retain the characteristic supramolecular motion of their homopolymer counterparts. These critical insights into supramolecular dynamics offer strategies to tailor the dynamic functions of supramolecular nanomaterials.

Inspired by natural protein assemblies[1,2], synthetic supramolecular polymers have emerged as a platform to design functional nanomaterials[3], including bioactive ones that closely mimic the structures, dynamics, and biofunctions of their natural analogs[4–6]. The tunable energy of dynamic non-covalent interactions among monomeric building blocks, relative to the covalent bonds that connect structural units in traditional polymers, opens opportunities to design responsive and adaptive systems with potential applications in environmental sustainability, optoelectronics, and medicine[3,4,7–10]. Moreover, supramolecular polymerization via self-assembly of monomers enables the facile incorporation of multiple functional components into a single synthetic dynamic system, thereby mimicking more closely the complex natural analogs[11–15]. Despite the growing recognition of the potential advantages of dynamic supramolecular polymers over their covalent counterparts, unraveling the nature of dynamics in both homopolymerization and multi-component copolymerization systems, particularly in aqueous environments, remains a significant challenge due to the assembly pathway complexity. This complexity often leads to the formation of assemblies that are in kinetically trapped states

[1]Center for Regenerative Nanomedicine, Northwestern University, Chicago, IL, USA. [2]Institute for Complex Molecular Systems, Eindhoven University of Technology, Eindhoven, The Netherlands. [3]Department of Chemistry and Chemical Engineering and Laboratory of Macromolecular and Organic Chemistry, Eindhoven University of Technology, Eindhoven, The Netherlands. [4]Department of Chemistry, Northwestern University, Evanston, IL, USA. [5]Department of Applied Physics and Science Education, Eindhoven University of Technology, Eindhoven, The Netherlands. [6]Department of Biomedical Engineering and Synthetic Biology Group, Eindhoven University of Technology, Eindhoven, The Netherlands. [7]School of Chemistry and RNA Institute, University of New South Wales, Sydney, NSW, Australia. [8]Department of Medicine, Northwestern University, Chicago, IL, USA. [9]Department of Biomedical Engineering, Northwestern University, Evanston, IL, USA. [10]Department of Materials Science and Engineering, Northwestern University, Evanston, IL, USA. ✉e-mail: e.w.meijer@tue.nl; s-stupp@northwestern.edu

rather than in thermodynamic equilibrium states encoded by molecular design[16–18].

To address this challenge, it is crucial to develop experimental methodologies that are capable of precisely probing in situ the self-assembly process of monomers under thermodynamic equilibrium during supramolecular polymerization. A seminal example is the cooling methodology, which monitors in situ temperature-induced assembly process under thermodynamic equilbrium[19,20]. Key principles in this methodology include (i) erasing any kinetic history by heating the system to depolymerize preformed polymers, (ii) subsequently inducing the formation of polymers under thermodynamic equilibrium by precisely and slowly cooling the system, and (iii) monitoring in situ the polymerization process during cooling using various characterization techniques, such as UV–Vis absorbance, circular dichroism (CD), or fluorescence spectroscopy, with complementary thermodynamic modeling. This combinatorial methodology has provided an indispensable basis in establishing the mechanistic framework of supramolecular polymerization over the past decades[20,21], such as nucleation/elongation mechanisms, asymmetric amplification rules, and copolymerization regimes. However, the cooling methodology is often unsuitable for aqueous systems, where strong hydrogen bonding or hydrophobicity of molecules complicate the assembly pathways during the heating and cooling processes[17]. For a variety of aqueous assemblies, pH titrations are employed to investigate the charged systems[6,22–32], since their assembly states can be conveniently controlled through deprotonation and protonation by adjusting the pH. While these studies have provided many important observations and insights, an important next step is to increase the level of precision using a combinatorial approach similar to the well-established cooling methodology. This approach would lead to a deeper understanding of supramolecular polymerization in aqueous systems, which are critically significant as they are closer analogs to biological assemblies.

Here, we introduce a general titration methodology to probe in situ the self-assembly process for aqueous systems by revisiting the pH titration experiments. Our approach involves a simple yet effective modification of conventional titration experimental setups, combined with different in situ and ex situ spectroscopy, microscopy, X-ray scattering, and thermodynamic modeling. This combinatorial approach enables the precise mapping of the self-assembly landscape of one- and two-component supramolecular polymerization systems under thermodynamic equilibrium, using peptide amphiphiles as classical model molecules. This comprehensive level of access provides critical insights into the binary assembly mechanism of PA supramolecular polymerization systems as well as the complex nature of supramolecular dynamics, as related to monomer structure, assembly equilibrium, internal structural order, and copolymerization regime (see Fig. 1).

## Results

### Probing the formation of PA supramolecular homopolymers

We chose three positively charged peptide amphiphiles (PA) as model molecules to develop the titration methodology (see Fig. 1). In an aqueous environment, these molecules are expected to form filamentous supramolecular polymers, as a result of their non-covalent interactions including hydrogen bonding that under certain conditions leads to β-sheet secondary structure, electrostatic repulsion among charged residues, as well as hydrophobic collapse of their alkyl tails[17]. Incorporation of biological signals in the peptide sequence of PA monomers is well known to create highly bioactive materials for regenerative medicine that closely mimic extracellular matrices[5,33,34]. In analogy to the cooling methodology, the titration methodology was implemented by first adding hydrochloric acid (HCl) to aqueous stock solutions of the cationic PAs to fully protonate the assemblies to maximize electrostatic repulsions to a point where the preformed

supramolecular filaments depolymerize into small aggregates. CD experiments on the PA molecule $C_{16}V_3A_3K_3$ (PA-1, Fig. 1b) reveal the presence of random coils with their characteristic peak at 195 nm at a total monomer concentration ([PA]) of 20 μM (Fig. 2a), in contrast to the CD spectra of β-sheet polymers typically found in PA systems[17]. These results indicate that, like common surfactants, PA monomers form random-coil micellar assemblies under this condition due to their intrinsic amphiphilicity, which is further supported by the CMC value of PA-1 at 1 μM obtained using the Nile Red assay (Supplementary Fig. 1). To prevent kinetic traps, the formation of supramolecular polymers was precisely controlled by subsequently titrating a sodium hydroxide (NaOH) solution into the PA micellar solution at well-defined flow rates using a syringe pump. The evolution of secondary structures formed by PA-1 was monitored in situ using CD spectroscopy (Fig. 2a and Supplementary Figs. 2–4). During the titration with NaOH, the resulting CD spectra show a shift in the secondary structure from random coils to β-sheets, characterized by a negative Cotton effect with two peaks at 203 nm and 220 nm. These results indicate that the NaOH added to the micellar solution reduces electrostatic repulsion among monomers and induces the formation of β-sheets, a characteristic secondary structure of PA-1 polymers as previously reported by Stupp laboratory[17]. Additionally, rate-dependent titration and thermal annealing CD measurements at 80 °C verify that the probed formation of β-sheets occurs under thermodynamic equilibrium conditions only at sufficiently low titration rates, for example 0.5 mL hour$^{-1}$ (Supplementary Figs. 5 and 6).

To quantify the formation of β-sheet polymers, we selectively monitored the molar ellipticity at 220 nm ($\theta_{220nm}$) during NaOH titration, yielding CD titration curves (Fig. 2b). Subsequent normalization of these curves provides the extent of polymerization ($\phi_{\beta-sheets}$), as the molar fraction of monomers contained within β-sheets as a function of the molar ratio of NaOH equivalents to total PA monomers ([NaOH]/[PA]) (Fig. 2c and Supplementary Fig. 7). These curves reveal that increasing [NaOH]/[PA] ratios above a critical value result in higher $\phi_{\beta-sheets}$, which can be attributed to reduced electrostatic repulsion among PA monomers as the positively charged lysine residues are deprotonated by the titrated NaOH. Testing the influence of [PA] on the β-sheet formation process, the titration curves and their normalizations at different [PA] indicate that higher values lead to more favorable formation of β-sheets at the same [NaOH]/[PA] (Supplementary Fig. 7a). This mirrors the previous observation that higher concentrations of charged PA monomers provide screening for repulsive electrostatic interactions[17]. Hence, by plotting $\phi_{\beta-sheets}$ as a function of [NaOH]/[PA] and [PA], the influence of these two polymerization conditions on the propensity of PA-1 molecules for self-assembly can be visualized in the form of an assembly landscape (Fig. 2d and Supplementary Figs. 7–16). A comparison of the assembly landscape of PA-1 with two counterparts containing altered peptide sequences in the β-sheet forming region, $C_{16}V_3G_3K_3$ (PA-2) and $C_{16}A_3G_3K_3$ (PA-3), reveals a decreasing trend in $\phi_{\beta-sheets}$ from PA-1 to PA-2 and PA-3 under the same polymerization conditions (Fig. 2d). This trend correlates strongly with the decreasing hydrophobicity of the side chains in the amino acids of the peptide sequences: valine (V, -CH(CH$_3$)$_2$) > alanine (A, -CH$_3$) > glycine (G, -H), which results in reduced β-sheet strength due to less effective shielding of hydrogen bonding within supramolecular polymers from the surrounding water, as well as a diminished hydrophobic collapse of monomers contributed by these side chains[24]. Although NaCl is also formed during the titration, its impact on PA polymerization via charge screening[17] is weaker than that of hydroxide ions, which directly modulate the protonation states of lysine, and can therefore be reasonably neglected in our analysis (Supplementary Figs. 15, 47, and 48).

The CD titration curves reveal two plateau regions, each corresponding to a distinct secondary structure in the PA assemblies: random coils or β-sheets. A combination of transmission electron

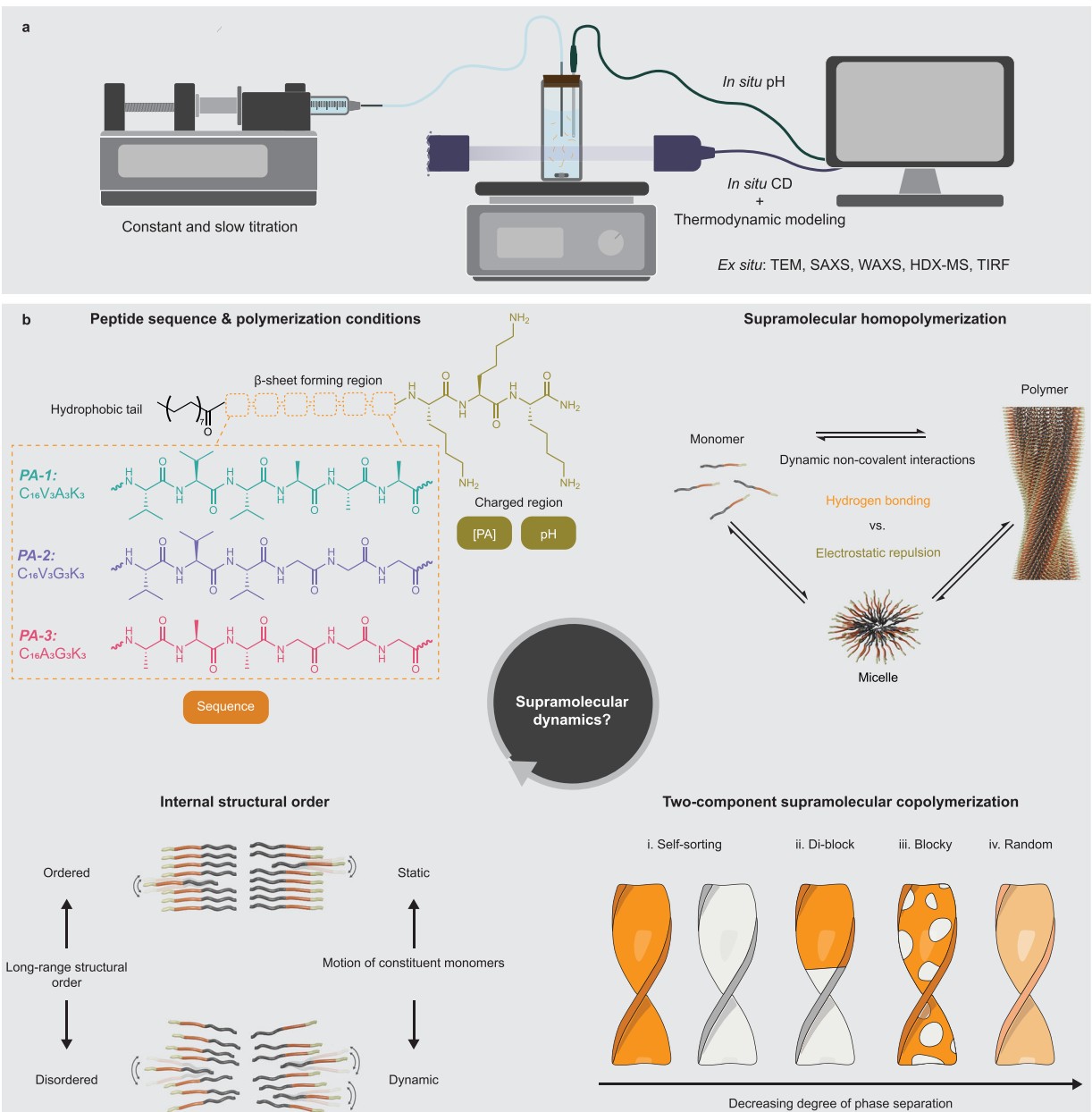

Fig. 1 | **Methodology and concept of this work. a, b** Schematic illustration of (**a**) the combinatorial base-titration methodologies and (**b**) the correlation between supramolecular dynamics and monomer structure, assembly equilibrium, internal structural order, as well as copolymerization regime within peptide amphiphile (PA) supramolecular polymerization model systems.

microscopy (TEM) (Fig. 2e and Supplementary Figs. 17 and 18), small angle X-ray scattering (SAXS) (Fig. 2h and Supplementary Fig. 19), and CD spectroscopy (Supplementary Fig. 20) indicates assemblies in the first plateau region, where peptide sequences adopt a random coil conformation, exclusively form spheroidal aggregates, while those in the second plateau region, characterized by β-sheet secondary structures, form nanoribbons. The absence of detectable $^1$H NMR signals further confirms the depletion of micelles in the second plateau (Supplementary Fig. 21b). From this point forward, we refer to the spheroidal aggregates as "micelles" and the filamentous assemblies as "polymers". Furthermore, the consistent normalizations of CD titration curves at different wavelengths (Supplementary Figs. 4 and 10) suggests the absence of a third assembled structure during the transformation between these two assemblies. This conclusion is

further supported by the consistent width of polymers extracted from TEM images (Fig. 2f and Supplementary Tables 6–8), indicative of the same monomer packing within PA polymers, and the constant diffusion coefficients of micelles measured with diffusion-ordered spectroscopy (DOSY) NMR (Fig. 2g and Supplementary Figs. 22 and 23), indicative of unchanged micelle size, during different stages of polymerization. Additionally, the relatively low CMC values of a few µM for the model PA molecules obtained from Nile Red assay (Supplementary Fig. 1) suggests that the presence of free monomers is negligible relative to the significantly larger total monomer concentration ([PA]), as confirmed by the constant characteristic molar ellipticity of micellar assemblies at different PA concentrations (Supplementary Fig. 24). Thus, our PA supramolecular polymerization system can be described as a binary assembly system composed of micelles with random coil

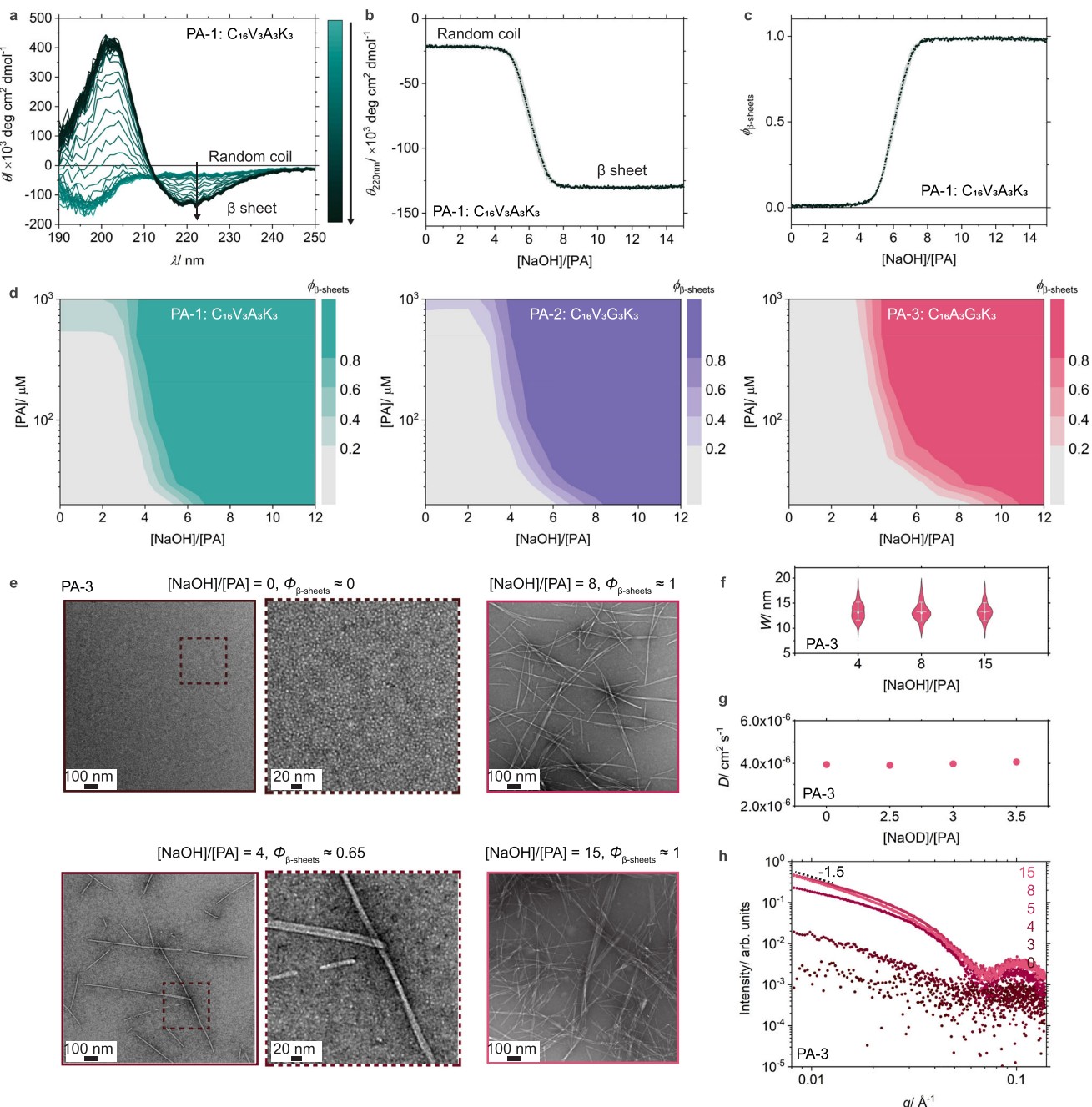

**Fig. 2 | Supramolecular homopolymerization of PA. a** Evolution of circular dichroism (CD) spectra measured for 20 μM PA-1 during the titration of NaOH solution. **b** Molar ellipticity at 220 nm ($\theta_{220nm}$) and **c** the molar fraction of monomers within β-sheets ($\phi_{\beta-sheets}$) as a function of the molar equivalents of titrated NaOH to total PA monomers ([NaOH]/[PA]). All CD titrations were repeated 5 times ($N = 5$). **d** Assembly landscapes for the three model PA systems at different concentrations of PA monomers ([PA]) and [NaOH]/[PA]. **e** Negative-staining transmission electron microscopy (TEM) images of the PA-3 polymerization system at different [NaOH]/[PA] ratios. **f** Violin plots with mean (lines) and median values (circles) of the polymer width ($W$) extracted from the TEM images. $N = 125$, 181, and 148 for [NaOH/[PA] = 4, 8, and 15. **g** Diffusion coefficients ($D$) of PA-3 micelles plotted as a function of [NaOD]/[PA] ratios. **h** Small-angle X-ray scattering (SAXS) profiles measured for the PA-3 polymerization system at different [NaOH]/[PA] ratios. Error bars denote the standard deviation (SD).

conformation and β-sheet-containing supramolecular polymers with well-defined structural characteristics. The titrated NaOH only modulates the relative populations of micelles and polymers, which can be precisely quantified using CD titration curves. Interestingly, when applying the same normalization procedure to $^1$H NMR results, $\phi_{\beta-sheets}$ values are consistent with those obtained from CD normalization (Supplementary Fig. 21c), further confirming the binary assembly nature of PA polymerization. Hence, our methodology allows us to uncover the binary assembly nature of PAs and provides precise

assembly landscapes to elucidate how peptide sequence and polymerization conditions influence the population of micelles and polymers during PA polymerization.

## Mechanism of the micelle-to-polymer transformation
Our results quantitatively show that titrated NaOH induces the transformation of micelles into polymers under thermodynamic equilibrium during PA polymerization. We can therefore apply a theoretical model to fit our data to gain a better understanding of this process. Drawing

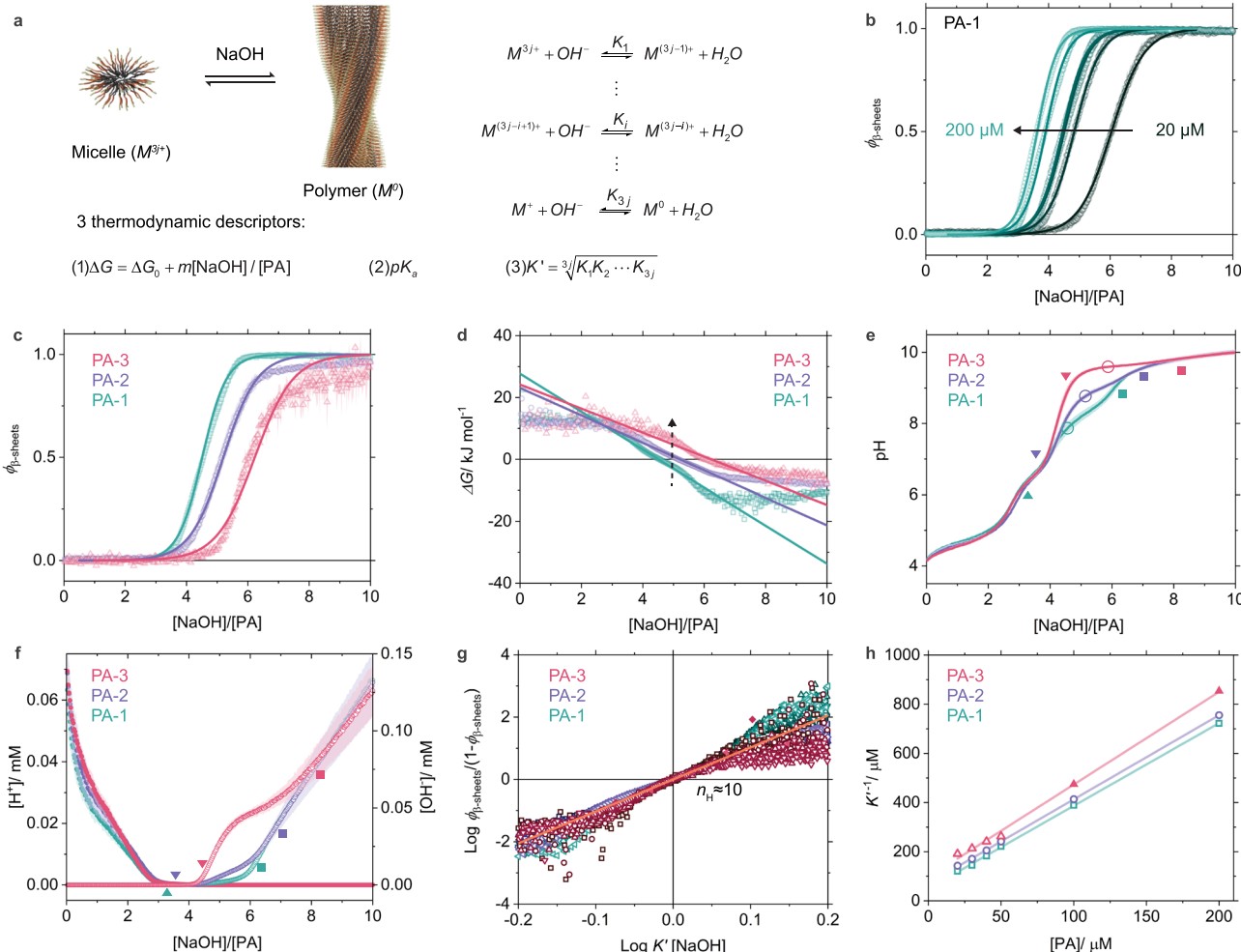

**Fig. 3 | Mechanism of the micelle-to-polymer transformation. a** Schematic overview of the equilibrium between micelles and polymers as well as three thermodynamic descriptors introduced to depict the transformation. **b** Fits (lines) of the normalized CD titration curves (symbols) of PA-1 at different [PA] with the *m*-factor model. **c, d** Comparison of experimentally obtained (symbols) and fitted (lines) (**c**) $\phi_{\beta-sheets}$ and (**d**) apparent Gibbs free energy change ($\Delta G$) of three PA systems at [PA] = 40 μM. **e** In situ pH titration curves of three PA systems at [PA] = 40 μM, $N = 4$. The apparent p$K_a$ values are indicated by hollow circles. In (**e**)

and (**f**), the onsets and ends of micelle-to-polymer transformation are marked with triangles and squares, respectively, based on CD titration curves in (**c**). **f** Concentrations of free protons ([H⁺]) and hydroxide ions ([OH⁻]) in the solution extracted from the pH titration curves. **g** Hill plot of all the three PA systems at various [PA] obtained by fitting the allosteric binding model. **h** Comparison of the apparent binding constants ($K'$) for the deprotonation of micelles at different [PA] extracted with the model. The linear fits of these data are shown in lines. Error bars denote the standard deviation (SD).

inspiration from the studies of protein unfolding[35] and solvent-induced supramolecular polymerization in organic solvents[36], we introduced a simple yet effective *m*-factor mass-balance model to describe this transformation process. In this model, the apparent relative Gibbs free energy change ($\Delta G$) for the transformation is assumed to be linearly dependent on the [NaOH]/[PA] molar ratio with a coefficient of *m* (Fig. 3a and Supplementary Text 1). Fitting the normalized CD titration curves in the model generates the values of $\Delta G$ for the three model PA polymerization systems under different conditions (Fig. 3b, c and Supplementary Fig. 25). This model shows that the thermodynamic driving force for supramolecular polymerization at a specific [NaOH]/[PA] molar ratio is the lowest for PA-3 and highest for PA-1 with intermediate values for PA-2 (Fig. 3d and Supplementary Tables 9–11).

Next, we monitored in situ the pH evolution during titration. The pH titration curves of the three PA polymerization systems reveal clear differences in their respective [NaOH]/[PA] ranges where the transformation of micelles into polymers occurs (Fig. 3e and Supplementary Figs. 26 and 27). An abrupt decrease in the slope of [OH⁻] curves accompanies the onset of polymerization, whereas [OH⁻] increases with

a nearly constant rate after the completion of polymerization, as exemplified in PA-3 assemblies (Fig. 3e, f). These results suggest that a substantial fraction of titrated hydroxide ions is consumed by the deprotonation of monomers in micelles in the transformation into supramolecular polymers. Interestingly, as we switched from PA-1 to PA-2 and then PA-3, the "buffering" effect in the pH titration curves and therefore the consumption of titrated hydroxide ions becomes less pronounced even with the same amount of titrated NaOH. The complexity of these pH titration curves (Fig. 3e) prevents us from directly extracting pKa values from the plateau regions, as done in previous studies on supramolecular gel systems[23–25]. Instead, we define the apparent p$K_a$ for each polymerization as the pH value measured at $\phi_{\beta-sheets} = 0.5$, corresponding to an equal fraction of protonated micelles and polymers consisting of deprotonated monomers. The pKa values at [PA] = 40 μM increase from 7.85 to 8.76 and 9.60 for PA-1, PA-2, and PA-3, respectively. All these values are substantially lower than that corresponding to free lysine (pKa = 10.5), implying that stronger cohesive hydrogen bonding and hydrophobic collapse of aliphatic segments can lead to more effective deprotonation of monomers. To model the

deprotonation process of micelles during transformation, we introduced an allosteric binding model to fit the normalized CD titration curves (Fig. 3a, Supplementary Text 2), by analogy to the cooperative ligand-receptor binding in biological signaling processes[37,38]. In this model, the micelles containing $j$ monomers ($M^{3j+}$) are considered as multivalent receptors with $3j$ binding sites and the NaOH is regarded as a monovalent ligand. The sequential deprotonation process is assumed to undergo via the ligand-receptor binding governed by the apparent binding constant ($K'$). Interestingly, the resulting Hill plots, representing $\log \phi_{\beta-\text{sheets}}/(1 - \phi_{\beta-\text{sheets}})$ as a function of $\log K'[NaOH]$, display similar Hill coefficients ($n_H$) of around 10, corresponding to the slope at the origin of axes for all the three PA systems (Fig. 3g and Supplementary Figs. 28 and 29). A Hill coefficient significantly greater than 1 indicates a cooperative deprotonation process of micelles[38]. It is important to note that the presence of a small amount of carbonate is not considered in the calculation of the Hill coefficients and this may have an impact on the calculated values; as a result, the deprotonation process of the peptide amphiphiles micelles inferred from the Hill analysis may be slightly under- or overestimated. Such cooperativity is expected to result in extensive deprotonation of a subset of micelles to form polymers with a negligible amount of intermediate charged states, while the remaining micelles stay fully charged. This mechanism is expected to facilitate a binary assembly model rather than a multistep assembly process, consistent with our experimental observations described above. Additionally, depending on the assumed extent of final deprotonation of micelles, the Hill coefficient of 10 suggests an estimated micelle size ranging from approximately 3 to 10 monomers. Based on this model, the tendency for deprotonation decreases as the apparent binding constant ($K'$) decreases from $5.5 \times 10^{-3} \, \mu M^{-1}$ in PA-1 to $4.9 \times 10^{-3} \, \mu M^{-1}$ in PA-2 and $4.1 \times 10^{-3} \, \mu M^{-1}$ in PA-3 at [PA] = 40 μM (Fig. 3h). These results suggest that despite the identical peptide sequence in the terminal charged domain of all three monomers, their differences in ability to form strongly bonded β-sheets directly affects deprotonation and polymerization propensity.

### Internal structures and dynamics

To assess the internal structures of homopolymers, we compared the CD spectra of the three PA samples primarily composed of β-sheet polymers, located at the second plateau region in the CD titration curves under the guidance of assembly landscape ($\phi_{\beta-\text{sheets}} \approx 1$). While all these polymers display the characteristic Cotton effect of β-sheets, both the peak intensity and wavelength are highest in PA-1, decrease in PA-2, and are lowest in PA-3 (Fig. 4a and Supplementary Fig. 30). This trend indicates a decreasing level of twisting for the helical stacking of monomers within the β-sheet PA polymers, consistent with the results for polypeptide folding[39]. Similarly, the intensity of the crystalline diffraction peaks around $1.4 \, \text{Å}^{-1}$ in the wide-angle X-ray scattering profiles, corresponding to the characteristic $d$-spacing of 4.5 Å for β-sheets, decreases with the same trend (Fig. 4b and Supplementary Fig. 31). This trend suggests an increasing degree of internal disorder within the β-sheet PA polymers, greatest in PA-3 assemblies with a larger average width ($W$) relative to those formed by PA-1 and PA-2 (Fig. 4c and Supplementary Fig. 32, Supplementary Tables 6–8). Therefore, by varying the hydrogen bonding strength associated with the peptide sequence of PA molecules, it is possible to tune the equilibrium between small micelles and polymers as well as the degree of internal order.

Next we employed hydrogen−deuterium exchange mass spectrometry (HDX-MS)[40,41], a technique not requiring the labeling of monomers with fluorophores[42] or spin labels[43], to probe the dynamics within the PA polymers ($\phi_{\beta-\text{sheets}} \approx 1$) (Supplementary Fig. 33). Despite the wide distribution of various exchanged species containing $n$ deuterium atoms (PA$n$D), the molar fraction of the fully exchanged PA17D isotope ($\phi_{\text{PA17D}}$) increases without a substantial increase of $\phi_{\text{PA15D}}$ and $\phi_{\text{PA16D}}$ during the exchange process of all three polymers (Fig. 4d–f and Supplementary Text 3). This result indicates that the primary factor

contributing to the formation of PA17D is the instant exposure of the entire monomer to ambient $D_2O$ due to its dynamic motion via either monomer exchange or micelle-to-polymer equilibrium rather than $D_2O$ diffusion into the interior of polymers (Fig. 4g). Therefore, the kinetics of PA17D isotope formation can serve as an effective indicator for the dynamics of polymers (Fig. 4h). These three PA polymers exhibit significant differences in their exchange kinetics, with $\phi_{\text{PA17D}}$ values after 96 h for PA-1, PA-2, and PA-3 of 16%, 51%, and 86%, respectively. This enhancement in exchange kinetics strongly indicates a correlation between low levels of internal order and greater supramolecular dynamics. Additionally, if the [NaOH]/[PA] ratio is kept at 0 such that PA-3 monomers mainly form micelles ($\phi_{\beta-\text{sheets}} \approx 0$), $\phi_{\text{PA17D}}$ rises rapidly to 95% within 15 min (Fig. 4h and Supplementary Fig. 34). Hence, PA micelles display even much faster dynamics than their polymer counterparts, indicating the dynamics of supramolecular polymerization systems is also influenced by the equilibrium between spheroidal micellar aggregates and the elongated polymers.

Interestingly, the good fit obtained for the exchange kinetics curves with a bi-exponential function rather than a mono-exponential one suggests that the exchange process occurs with at least two different exchange rate constants: $k_{\text{fast}}$ and $k_{\text{slow}}$ (Table 1, Supplementary Fig. 35, and Supplementary Tables 13–14). The increasing trend in the exchange rate constants from PA-1 to PA-2 and PA-3 confirms the trend of enhanced dynamics in these three PA polymers. Moreover, the two rate constants imply a heterogeneity in internal structure in each PA polymer. PA monomers in ordered domains within polymers interact more strongly with adjacent ones and therefore display slower exchange, whereas other monomers are weakly bonded and therefore exchange more rapidly.

### Two-component supramolecular copolymerization of PA

With the methodology described above and detailed knowledge of the homopolymerization, we then investigated the copolymerization. For this purpose, we probed in situ the polymerization process of two-component monomer mixtures with varying degrees of monomer excess (*m.e.*) defined as the difference in mole fraction between that of the monomer present in excess and the co-monomer. The CD titration curves of the mixtures composed of PA-1 and PA-2 (PA-1/2) monomers exhibit a similar shape as for the homopolymerization curves, while those of PA-1/3 and PA-2/3 systems display two distinct polymerization regimes (Fig. 5a and Supplementary Fig. 37). To study the phase separation within copolymers, we extracted the CD intensity of copolymerization systems at the second plateau region in the titration curves ($\phi_{\beta-\text{sheets}} \approx 1$) and plotted them as a function of *m.e.*. We observed that the CD intensity of PA-1/3 and PA-2/3 systems resemble the linear combination of their respective homopolymers, whereas that of the PA-1/2 system substantially deviates from the linear combination (Fig. 5b and Supplementary Fig. 38). The nonlinear dependence of copolymer properties on the composition is often used as an effective indicator for the degree of component mixing[32,44]. In line with this principle, our results suggest that PA-1 and PA-2 monomers are well mixed, whereas PA-1/PA-3 and PA-2/PA-3 monomers are preferentially phase-separated on the supramolecular scale. Additionally, the apparent pKa values for all three copolymerization systems fall between those of their respective homopolymerization counterparts (Supplementary Fig. 39), indicating that copolymerization influences the deprotonation process of micelles.

To investigate the phase separation of monomers at the individual supramolecular polymer scale, we covalently labeled monomers with the fluorescent dyes Cy3 and Cy5 and visualized their spatial distribution within copolymers formed with equimolar monomers (*m.e.* = 0%) using total internal reflection fluorescence (TIRF) microscopy[42,45,46]. In all three copolymerization systems, both monomers co-exist within the same polymers as short segments, without any obvious self-sorting into separate polymers or large di-block microstructures (Figs. 1b, and 5e and Supplementary Figs. 40–45). These results suggest the formation of blocky supramolecular polymers

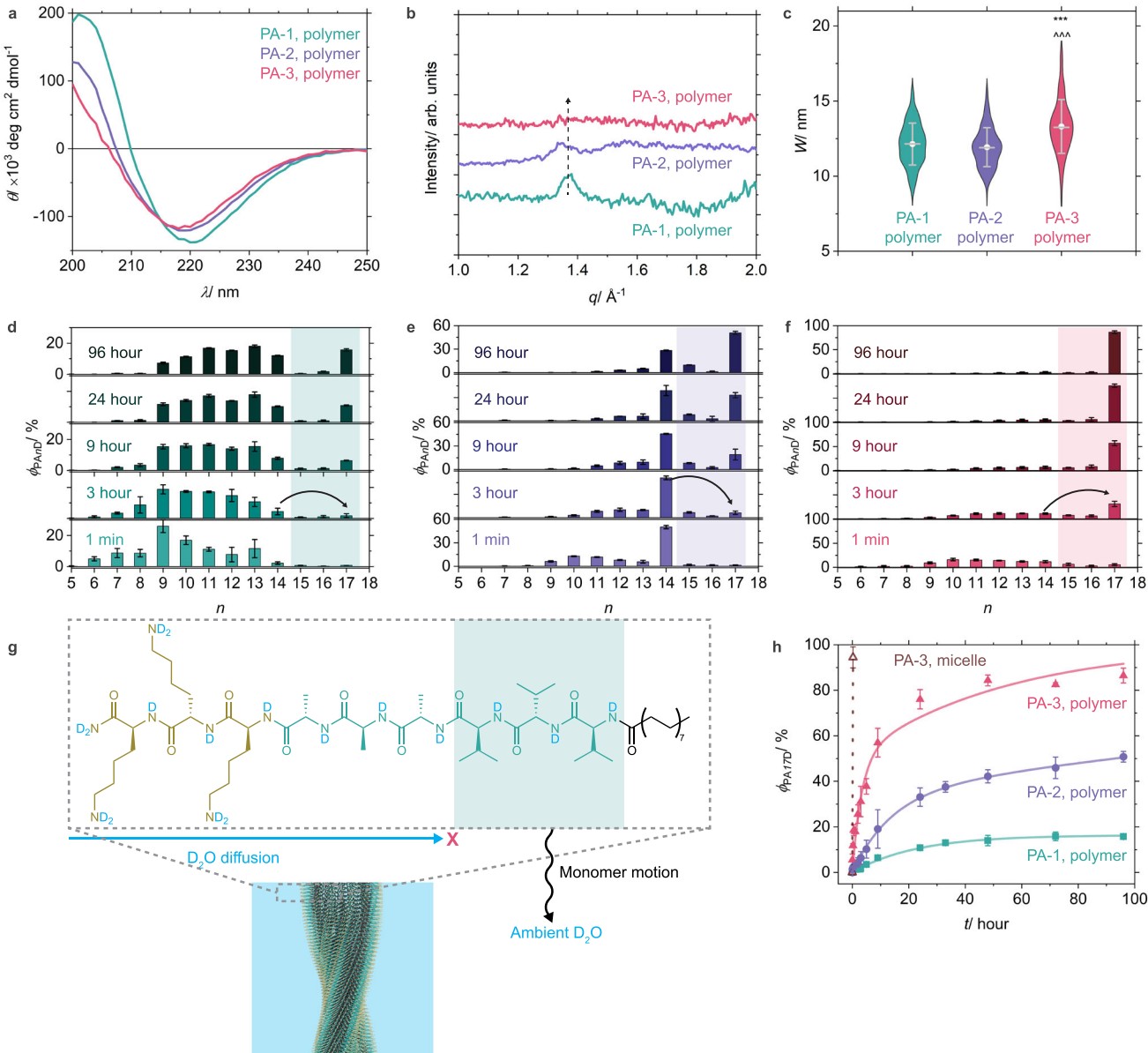

**Fig. 4 | Internal structure and dynamics of PA homopolymers. a** CD spectra, **b** Wide-angle X-ray scattering (WAXS) patterns, and **c** Violin plots with mean (lines) and median values (circles) of the width ($W$) of three PA homopolymers formed at [NaOH]/[PA] = 15. ***$P < 0.001$ PA-3 vs. PA-1, ^^^ $P < 0.001$ PA-3 vs. PA-2; one-way analysis of variance (ANOVA) with a Tukey's multiple comparison test, $\alpha = 0.05$. [PA] = 500 μM for (**a**) and (**c**). [PA] = 2 mM for (**b**). $N$ for PA-1, PA-2, and PA-3 is 174, 193, and 148, respectively. **d**–**f** Distribution of various hydrogen/deuterium (H/D) exchanged PA species (PA$n$D) containing $n$ deuterium atoms at various exchange time ($t$) for (**d**) PA-1, (**e**) PA-2, and (**f**) PA-3 polymers. Measurements were taken from distinct samples ($N = 3$). **g** Illustration of the H/D exchange mechanism for the formation of fully exchanged PA17D isotope of PA-1. **h** H/D exchange kinetics of three PA polymers and PA-3 micelles ([NaOH]/[PA] = 0). Bi-exponential fits of the experimental kinetic data are shown in solid lines. Error bars denote the standard deviation (SD).

containing multiple blocks of each monomer. To further differentiate the degrees of internal phase separation (or blockiness) in these blocky copolymers, as suggested by the CD titration results, we conducted a high-throughput cross-correlation analysis of the collected TIRF

**Table 1 | Exchange rate constants and molar fractions of two domains in the homopolymers derived from bi-exponential fits of the H/D exchange kinetics shown in Fig. 4h**

| Polymer | $k_{fast}$/ hour$^{-1}$ | $\phi_{fast}$/ % | $k_{slow}$/ hour$^{-1}$ | $\phi_{slow}$/ % |
|---|---|---|---|---|
| PA-1 | $4.4 \times 10^{-2}$ | 16.4 | $<1.0 \times 10^{-5}$ | 83.6 |
| PA-2 | $7.3 \times 10^{-2}$ | 34.2 | $3.0 \times 10^{-3}$ | 65.8 |
| PA-3 | $3.0 \times 10^{-1}$ | 50.4 | $1.9 \times 10^{-2}$ | 49.6 |

images. This analysis reveals a wide distribution of the normalized cross-correlation coefficients (NCC) between Cy3 and Cy5 channels at the individual pixel scale (160 nm × 160 nm), suggesting highly diverse degrees of internal phase separation within all three copolymers (Fig. 5e and Supplementary Fig. 46). However, this large variation makes it challenging to extract a reliable representative characteristic index using traditional statistical metrics, such as mean values with standard deviation or median values. Additionally, when we switched the cyanine labels between the two monomers within the same system, such as Cy3-PA-1/Cy5-PA-2 versus Cy5-PA-1/Cy3-PA-2, both the distribution and average values of NCC varied significantly. This inconsistency is likely due to the subtle structural difference between Cy3 and Cy5 dyes, which may alter the delicate balance of electrostatic repulsions, hydrogen bonding, and hydrophobicity of the labeled PA

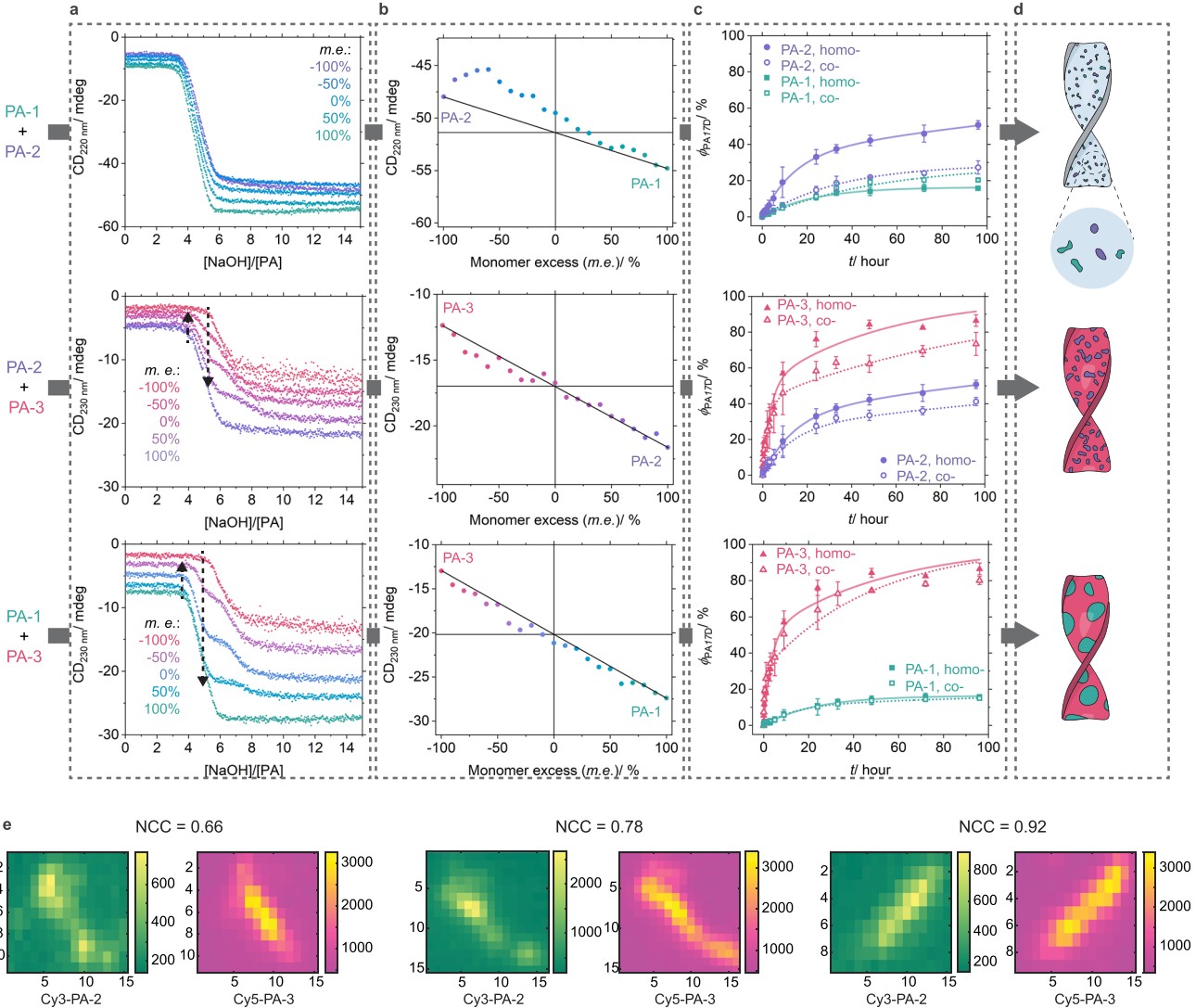

**Fig. 5 | Supramolecular copolymerization of PA. a** Representative CD titration curves of supramolecular copolymerization of PA-1/2, PA-2/3, and PA-1/3 systems measured at different monomer excess (*m.e.*), while [PA]$_{total}$ is maintained at 40 μM. The onsets of two polymerization regimes are marked with arrows. **b** CD intensity of copolymers extracted at [NaOH]/[PA] = 12 as a function of *m.e.*. The linear combination of the CD intensity of respective homopolymers is indicated by the black line. **c** H/D exchange kinetics of PA copolymers (hollow symbols) formed at *m.e.* = 0% and their respective homopolymers (solid symbols). Bi-exponential fits of the experimental kinetic data are shown by solid lines for homopolymers and dashed lines for copolymers. Measurements were taken from distinct samples

(*N* = 3). **d** Schemes for the proposed microstructures of two-component PA supramolecular copolymers with increasing degrees of internal phase separation. **e** Representative total internal reflection fluorescence (TIRF) microscopy images taken in Cy-3 and Cy-5 channels using PA-2/3 copolymers as an example, which display various normalized cross-correlation coefficients (NCC) between these two channels at the individual pixel scale (160 nm × 160 nm). The unit on both axes in TIRF images is the pixel. 5 mol% of PA-2 monomers are labelled with Cy3 dyes and 5 mol% of PA-3 monomers with Cy5 dyes. *m.e.* = 0%. Error bars denote the standard deviation (SD).

monomers, though they are positioned far from the β-sheet peptide sequence to minimize the effects on the polymerization. Consequently, these issues prevent us from directly quantifying the degrees of phase separation within three blocky copolymers using cross-correlation analysis of TIRF microscopy images.

Given the challenge in the quantitative analysis of TIRF microscopy images, we employed HDX-MS to differentiate the degrees of internal phase separation at the molecular scale and to probe their correlation with supramolecular dynamics. We measured the exchange kinetics of three two-component copolymers (*m.e.* = 0%) and compared them with their respective homopolymers (Fig. 5c, Supplementary Fig. 36 and Supplementary Tables 15–17). Interestingly, for the well-mixed PA-1/2 copolymers, the exchange kinetics becomes faster for PA-1 monomers and slower for PA-2 monomers compared to

their respective homopolymers. Moreover, the exchange kinetics of these two monomers closely resemble each other, indicating they both have a similar molecular environment. The slight difference in their exchange kinetics suggests the presence of a small fraction of PA-1 or PA-2 blocks within co-assemblies of these two monomers. In contrast, the exchange kinetics of monomers in the PA-1/3 and PA-2/3 copolymers is slower than that measured in their respective homopolymers. Importantly, the reduction in dynamics observed for PA-3 monomers in PA-2/3 is larger than that in PA-1/3, implying the presence of a larger extent of phase separation within the PA-1/3 copolymers. Thus, by integrating HDX-MS measurements with CD and TIRF results, we establish a potent toolkit to resolve subtle differences in the degree of internal phase separation within our PA blocky copolymers, as schematically summarized in Fig. 5d.

## Discussion

The combinatorial titration methodology introduced here provides an unprecedented level of experimental access to the assembly mechanism and landscape, as well as its correlation with supramolecular dynamics in aqueous supramolecular polymerization systems, using peptide amphiphiles (PA) as models. Interestingly, this approach reveals a pH-dependent binary assembly equilibrium between spheroidal micelles and β-sheet polymers due to the cooperative deprotonation process of micelles[38]. In future work, it will be important to establish how the nature of peptide sequences leading to specific interactions among the monomers can change assembly polymorphism in supramolecular PA systems. Moreover, while NaOH was primarily used to induce polymerization here, we emphasize that this versatile methodology can be extended to study other polymerization conditions, such as NaCl and phosphate-buffered saline, which are frequently employed to trigger supramolecular polymerization. Our methodology displays the potential to precisely differentiate the distinct effects of these different polymerization conditions, as demonstrated in Supplementary Figs. 47 and 48, and even analyze their combined effects.

By integrating thermodynamic modelling into the CD titration curves, we derived a comprehensive set of quantitative descriptors that reveal how non-covalent interactions, including hydrogen bonding and electrostatic repulsion, influence the tendency of monomeric building blocks to assemble into supramolecular polymers, information that has been difficult to obtain in aqueous systems. These descriptors include the deprotonation capability of small micelles ($K'$) and the thermodynamic driving forces of their polymerization ($\Delta G$). Given the generality of these non-covalent interactions in governing self-assembly in synthetic and natural assembling systems, we envision that the thermodynamic descriptors enabled by our methodology will be highly beneficial in the current trend of data-driven discovery of functional nanomaterials[47,48]. Furthermore, quantitative HDX-MS measurements reveal the distinct dynamics exhibited by micelles and polymers coexisting in equilibrium, suggesting their cooperative roles in defining the overall dynamic functions of the supramolecular polymerization system. The structural and dynamic heterogeneity observed in PA polymers mirrors that found in natural ones such as disordered domains in the ordered tubulin lattice of microtubules, which are crucial for the reorganization of the cell cytoskeleton[49,50]. This example highlights the importance of understanding structural and dynamic heterogeneity to tailor the properties and functions of supramolecular polymers.

Our approach also surprisingly identifies that even a slight difference in peptide sequence mismatch can lead to varying degrees of internal phase separation and dynamic behavior in the resulting PA blocky copolymers. This resembles the phase separation found in cell membranes, where ordered clusters are embedded in disordered and dynamic lipid matrices due to molecular packing mismatch[51]. Such phase separation facilitates receptor clustering that plays a crucial role for the effectiveness of signaling pathways[51,52]. In analogy with lipid systems, when PA-3 monomers are copolymerized with PA-1 or PA-2, the lower tendency of their peptide sequence to engage in hydrogen bonding results in the formation of blocks with more ordered PA-1 or PA-2 monomers within a disordered and dynamic PA-3 matrix. Further phase separation into di-block supramolecular polymers or full self-sorting may be hindered by a favorable entropy of mixing in these structures as well as the reduced exchange dynamics at the block boundaries. A notable example recently reported from one of our laboratories demonstrated that the copolymerization of two PA monomers bearing different biological signals can significantly increase the bioactivity of supramolecular polymers for regenerative medicine, leading to enhanced repair of spinal cord injury in mice[34]. Motivated by this prominent example, the blocky structures within copolymers, along with their implications for spatial control over

clustering and dynamic control over the motion of constituent building blocks, opens intriguing possibilities for discovering emerging functions in multi-component supramolecular copolymers. Thus, we conclude that our combinatorial titration methodology provides a foundational framework for studying the complex dynamic nature of aqueous supramolecular polymers with a broad applicability to other assembly systems. The critical insights gained here enable strategies in designing supramolecular nanomaterials with tailored dynamic functions that are inaccessible in covalent polymers.

## Methods

### Synthesis of peptide amphiphiles (PAs)

Peptide amphiphiles $C_{16}V_3A_3K_3$ (PA-1), $C_{16}V_3G_3K_3$ (PA-2), and $C_{16}A_3G_3K_3$ (PA-3) were synthesized with standard fluoren-9-ylmethoxycarbonyl (Fmoc) solid-phase peptide synthesis on 4-methylbenzhydrylamine rink amide resin, using a CEM Liberty Blue automated microwave peptide synthesizer. Resin, Fmoc-protected amino acids and other solid-phase peptide synthesis reagents were purchased from Novabiochem (USA). Water was purified on an EMD Milipore Milli-Q Integral Water Purification System. Automated coupling reactions were conducted with 4 equivalents of Fmoc-protected amino acid or palmitic acid, 4 equivalents of N,N'-diisopropylcarbodiimide (DIC), and 8 equivalents of ethyl(hydroxyimino)cyanoacetate (Oxyma pure). Fmoc removal was achieved with 20% 4-methylpiperidine in dimethylformamide (DMF). Synthesized PAs were cleaved from the resin using a standard solution of 95% trifluoroacetic acid (TFA), 2.5% triisopropylsilane, and 2.5% $H_2O$. After concentrating the cleavage solution by rotary evaporation, PA products were precipitated in cold diethyl ether and dried under reduced pressure. The precipitates were dissolved in water ($\sim$10 mg ml$^{-1}$) containing 0.1 vol% TFA and purified through preparative-scale reversed-phase high-performance liquid chromatography (HPLC, Shimadzu Prominence), using an acetonitrile/water gradient containing 0.1 vol% TFA. Eluting fractions containing the targeted PAs were confirmed by mass spectrometry (Agilent 6520 QTOF LCMS). Acetonitrile was removed by rotary evaporation and PA products were subsequently lyophilized. The purity of lyophilized products was tested by LCMS (>95%), as shown in Supplementary Figs. 49–51.

For the PAs labeled with Cyanine3 (Cy3-) and Cyanine5 (Cy5-), Fmoc-azidolysine (Kaz) was coupled to the C-termini of the sequences to yield $C_{16}V_3A_3K_3$-Kaz, $C_{16}V_3G_3K_3$-Kaz, and $C_{16}A_3G_3K_3$-Kaz. These PAs were purified by HPLC. Purified PAs in slight molar excess were subsequently dissolved in DMF with either Cy3-DBCO or Cy5-DBCO (Vector Laboratories). The reaction was followed by mass spectrometry until the fluorescent starting material peak was consumed. Reaction mixtures were then purified by preparative HPLC and lyophilized. The purity of lyophilized products was tested by LCMS (>95%), as shown in Supplementary Figs. 52–57.

### Preparation of PA samples

To prepare the stock solution for homopolymerization experiments, PA powder was directly dissolved in a 10 mM HCl aqueous solution (2.5 equivalents) (Titripur®, Sigma-Aldrich) with a final monomer concentration ([PA]) of 4 mM, which ensured the depolymerization of any pre-formed PA polymers. By contrast, for the stock solution of fluorescently labeled PAs, 5 mol% of either Cy3-PA or Cy5-PA molecules were first mixed with 95 mol% their non-fluorescent counterparts in hexafluoro-2-propanol (HFIP) at 5 mg ml$^{-1}$ to achieve a homogeneous mixture. Similarly, for the stock solution of copolymerization systems, two PA monomers, with or without their fluorescent counterparts, were mixed in HFIP. These solution mixtures were dried under $N_2$ gas flow and then under vacuum. An aqueous solution of HCl (2.5 equivalents) was added to the dry mixtures to prepare PA stock solutions with targeted concentrations. All these stock solutions were vigorously mixed using a vortex mixer and ultrasonic bath, followed by

an aging process at room temperature overnight. To meet the requirements of different characterizations used in this work, PA stock solutions were diluted in Milli-Q water to a series of concentrations, as summarized in Supplementary Table 1. Diluted PA solutions were further treated using the same mixing and aging procedures prior to any subsequent experiments.

To prepare PA samples used for ex-situ characterizations, different equivalents of NaOH were added to the diluted PA solutions (containing 2.5 equivalents of HCl) to vary the population of micelles and polymers. After thorough mixing with a vortex mixer and ultrasonic bath, these PA samples were aged at room temperature for 7 days prior to different characterizations.

### In situ titration

2 ml of diluted PA solutions were added to a quartz cuvette of 10 mm path length (Hellma). During the titration experiments, 0.75 ml of NaOH aqueous solutions (Titripur®, Sigma-Aldrich) were constantly injected into the well-sealed cuvette with precisely regulated flow rates using a syringe pump (Harvard Apparatus Standard Infuse/Withdraw PHD Ultra Syringe Pump) under well stirring. The chosen concentrations of PA and respective NaOH are listed in Supplementary Table 1 to ensure the same dilution effect caused by the titration, as discussed in Supplementary Fig. 2. The evolution of CD spectra was recorded in situ using a Jasco J-815 spectropolarimeter. Similarly, the variation of pH values was monitored using a FiveEasy™ Plus FP20 pH meter equipped with InLab® Micro electrode and EasyDirect pH software (Mettler Toledo). For the in situ titration experiments of copolymerization systems, to minimize random errors introduced by weighing powders during the preparation of a series of stock solutions, the diluted PA micellular solutions ([PA] = 40 μM) of each molecule were directly mixed using a balance to prepare copolymerization solutions with various monomer excess (*m.e.*).

### Nile Red assay

The critical micelle concentration (CMC) of PA molecules in an aqueous environment was probed using Nile Red[46,53]. A stock solution of 1 mM Nile Red (Sigma-Aldrich) was prepared in ethanol (Biosolve). The 4 mM PA stock solution containing 2.5 equivalents of HCl was diluted in a series of twofold dilutions to prepare a range of PA concentration ([PA]). The Nile Red stock solution was freshly diluted 200 times in water to obtain a final concentration of 5 μM. Equal volumes of diluted Nile Red solution and PA solutions were subsequently mixed for each PA concentration. As a reference, an equal volume of water was added to the diluted Nile Red solution. All these solutions were aged in a 96 well microplates for 24 h to ensure complete depolymerization of PA polymers at concentrations below CMC. Fluorescence emission spectra were recorded using a Tecan FP Spark plate reader with an excitation wavelength of 550 nm and an emission range between 580 and 700 nm. The emission spectra of PA samples and water reference were averaged with 5 replicates and smoothed using the Savitzky-Golay function at a smoothing factor of 40. The peak position for each [PA] was extracted from its respective emission curve and the blue shift ($\Delta\lambda$) of the peak position was calculated as, $\Delta\lambda = \lambda_{peak, water} - \lambda_{peak, PA}$.

### $^1$H Nuclear Magnetic Resonance (NMR) spectroscopy

$^1$H NMR spectra were recorded on a Bruker-AVANCE III 500 MHz spectrometer equipped with a Prodigy BB cryoprobe at 298.15 K, with a 1H frequency of 500.13 MHz. Chemical shifts are reported in ppm relative to the residual peak of 3-(trimethylsilyl)propionic-*2,2,3,3-d*4 acid (TMS) in deuterium oxide ($D_2O$) solvent (Sigma-Aldrich). Two-dimensional (2D) Diffusion Ordered Spectroscopy (DOSY) spectra were acquired using the Longitudinal eddy current delay pulse program equipped with bipolar gradients pulse pair and 2 spoil gradients (ledbpgp2s). The DOSY maps were generated using the automatic processing option and the peak fit method with a single exponential

decay in MestReNova. To more accurately extract diffusion coefficients, the integrals of peaks corresponding to the methylene protons adjacent to the amine groups in the lysine residues were also fitted. The NMR sample preparation followed similar procedures for other ex-situ characterizations. A 1 mM stock solution of PA was prepared in $D_2O$ containing 2.5 equivalents of deuterium chloride (DCl, Sigma-Aldrich). Dry dimethyl sulfoxide (DMSO, Extra dry, Biosolve) at 1 mM was also added to the stock solution as an internal reference for peak integration. To prepare the assembly samples containing varying micelle fractions, different equivalents of sodium deuteroxide (NaOD, Sigma-Aldrich) were titrated to the stock solution.

### Negative-staining transmission electron microscopy (TEM)

TEM imaging was performed on a JEOL 1230 microscope at an accelerating voltage of 100 kV equipped with a Gatan 831 CCD camera. A droplet of 5–7 μl aqueous solution containing PA samples formed with different [NaOH]/[PA] at [PA] = 500 μM was deposited on the shiny side of a carbon film supported copper grid (300-mesh, Electron Microscopy Sciences) for 3 min. The sample was subsequently rinsed twice with water, stained with 2 wt% uranyl acetate, and dried in air for 10 min. The polymer width ($W$) was measured at the widest region of each polymer nanoribbon using the open-source ImageJ Fiji (Fiji is just ImageJ) platform[54]. Data was analyzed using GraphPad Prism software (version 5.04). Comparisons between pairs of experimental groups were performed using Kolmogorov–Smirnov test. Comparisons among three or more groups were conducted using ANOVA with Tukey's Multiple Comparison test.

### Synchrotron X-ray scattering

Experiments were performed at the DuPont-Northwestern-Dow Collaborative Access Team (DND-CAT) 5-ID-D station at the Advanced Photon Source (APS), Argonne National Laboratory[55]. Data were collected on a triple area detector system with an X-ray energy at 17 keV. The wavevector $q$ is defined as $(4\pi/\lambda)\cdot\sin(\theta/2)$, where $\theta$ is the total scattering angle. Samples were either measured in a flow-cell setup or a plate set-up. In the flow-cell setup, the PA samples were oscillated with a syringe pump during exposure to prevent beam damage. In the plate set-up, each sample was loaded in home-built sample cells comprising of two 30 μm thick AS32eco ultrathin glass (Schott AG) as window materials, attached to both sides of a 2.0 mm thick acrylic plate using 9474LE double-stick sheets (3 M) as adhesives. The acrylic plates and the double-stick sheets were laser-cut to form sample cells with 6 mm height and 3 mm width, and 36–38 μl of the solution was used for each cell. The sealed samples were then mounted on a translational stage at the beamline. Background samples containing water were also collected to perform background subtraction. The exposure time was 5 s or longer for the diluted samples to get sufficient signal-to-noise ratios. The data were averaged based on 5–10 frames. The acquired 2D scattering data were then reduced to 1D intensity vs. wavevector plots via azimuthal integration around the beam center in GSAS-II software[56] and were subtracted against water scattering profile before the analysis on power law of small angle and Bragg peaks at wide angles.

### Hydrogen/deuterium exchange mass spectrometry (HDX-MS)

4 mM PA polymer or micelle samples were diluted with $D_2O$ (Merck) 100 times. The diluted samples were stored at room temperature during the entire measurement. Aliquots of the diluted PA solutions were taken at specific time points and were subjected to electrospray ionization mass spectrometry (ESI-MS) to record their MS spectra. To facilitate the detection of monomers within the PA assemblies while minimizing artificial H/D exchange during the ESI-MS measurement, a PA aliquot was mixed with an equal volume of cold acetonitrile (−20 °C, Biosolve) containing 0.1 vol% formic acid (Sigma-Aldrich). The mixture was then injected into the mass spectrometer using a syringe

pump (Harvard Apparatus 11 Plus) at a flow rate of 50 µl min$^{-1}$. ESI-MS measurements were carried out using a Xevo G2 QTof mass spectrometer (Waters) with a capillary voltage of 2.7 kV and a cone voltage of 20 V. The source temperature was set at 100 °C, the desolvation temperature at 400 °C and the gas flow at 500 l h$^{-1}$. Among PA molecules with different charged states, the doubly charged ions, $[PAnD + 2D]^{2+}$ or abbreviated as PA$n$D, were the most pronounced species and therefore chosen for the quantitative analysis. The contribution from water and natural abundance of elements of other exchange species were deconvoluted from the original MS signal of PA$n$D using the Matlab® function lsqnonneg with the prefactors listed in Supplementary Table 12.

### Total internal reflection fluorescence (TIRF) microscopy

Eight-well chamber slides (ibidi µ-Slide) were successively cleaned by 2% sodium dodecyl sulfate aqueous solution, ethanol, and water. Slides were then treated with an aqueous solution containing 1 M NaOH for 1 h, followed by washing with water three times. To image the PA polymers, 50 µl of 40 µM PA solutions were freshly diluted to 10 µM in plastic microtubes using 150 µl 1.6 mM NaOH solutions, which can effectively prevent the dilution-induced depolymerization during the sample preparation and imaging. After the diluted PA samples were injected into the wells, 400 µl of 1 M NaCl solution was also added to each well to immobilize the polymer fibers onto the surface of glass substrates[46]. TIRF images were acquired on a Nikon N-STORM system using a Nikon 100 ×, 1.4NA oil immersion objective and a quad-band pass dichroic filter (97335 Nikon). Cy3- and Cy5-labeled samples were excited by 561 and 647 nm laser lines coupled into a single mode fiber by Agilent MLC400B laser combiner. All images were recorded with a size of 256 × 256 pixel (pixel size 0.17 µm) of an EMCCD camera (ixon3, Andor). Laser power, camera exposure time, readout speed and EM gain value were adjusted to allow comparable signal to noise ratio (SNR) in all images for cross-correlation analysis.

### High-throughput cross-correlation analysis

A series of image files (ND2), each corresponding to a distinct area on the sample coverslip with immobilized polymer fibers, were captured for further analysis. The images were acquired in two channels, Cy3 and Cy5, capturing different monomer components within PA copolymers. To extract information of polymer fibers from the acquired images, a robust Region of Interest (ROI) identification process was employed. We used adaptive thresholding to segment the images and identify potential ROIs to account for the non-uniform intensity profile of the different fibers. Key parameters such as size and the eccentricity of polymer fibers were optimized to filter ROIs based on thresholded binary image, ensuring relevance to polymer fiber structures. Cross-correlation analysis was then performed on corresponding ROIs from Cy3 and Cy5 channels to characterize the degree of colocalization of these two channels. The script employed the NCC at zero-shift position to quantify the similarity between the intensity patterns of ROIs in different channels. Statistical measures, including histograms of correlation coefficients, were generated to summarize the correlation distribution across the dataset. The combination of thresholding, cross-correlation, and statistical analyses, guided by optimized parameters, provided a robust methodology for extracting valuable information about the spatial relationships and co-localization of monomers within polymer fibers in the TIRF microscopy images.

## Data availability

The authors declare that the data supporting the findings of this study are available within this Article and its Supplementary Information. The Source Data of the main paper are provided with this Article. The Source Data of the Supplementary Information are available from the corresponding authors upon request. Source data are provided with this paper.

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

## Acknowledgements

We would like to thank Mark Richard Karver for his technical support on peptide synthesis, and Dr. Nuwanthika Kumarage and Dr. Mathijs Mabesoone for their technical assistance with NMR experiments. We also thank Prof. Anja Palmans, Dr. Lu Su, Dr. Giulia Lavarda, Dr. Shikha Dhiman, Prof. Yao Lin, and Dr. Bingbing Sun for helpful discussions. The peptide synthesis was performed at the Peptide Synthesis Core Facility of the Simpson Querrey Institute for BioNanotechnology at Northwestern University. This facility has current support from the Soft and Hybrid Nanotechnology Experimental (SHyNE) Resource (NSF ECCS-2025633). The Simpson Querrey Institute for BioNanotechnology, Northwestern University Office for Research, U.S. Army Research Office, and the U.S. Army Medical Research and Materiel Command have also provided funding to develop this facility. The X-ray scattering experiments were performed at the DuPont-Northwestern-Dow Collaborative Access Team (DND-CAT) located at Sector 5 of the Advanced Photon Source (APS). DND-CAT is supported by Northwestern University, The Dow Chemical Company, and DuPont de Nemours, Inc. This research used resources of the Advanced Photon Source, a U.S. Department of Energy (DOE) Office of Science User Facility operated for the DOE Office of Science by Argonne National Laboratory under Contract No. DE-AC02-06CH11357. Preliminary X-ray scattering data were also obtained at the LiX beamline at the National Synchrotron Light Source II, Brookhaven National Laboratory. The LiX beamline is part of the Center for BioMolecular Structure (CBMS), which is primarily supported by the National Institutes of Health, National Institute of General Medical Sciences (NIGMS) through a P30 Grant (P30GM133893), and by the DOE Office of Biological and Environmental Research (KP1605010). LiX also received additional support from NIH Grant S10 OD012331. As part of NSLS-II, a national user facility at Brookhaven National Laboratory, work performed at the CBMS is supported in part by the U.S. DOE, Office of Science, Office of Basic Energy Sciences Program under contract number DE-SC0012704. The TEM imaging made use of the BioCryo facility of Northwestern University's NUANCE Center, which has received support from the SHyNE Resource (NSF ECCS-2025633), the IIN, and

Northwestern's MRSEC program (NSF DMR-2308691). This work was partially supported by the European Research Council (SYNMAT project, ID 788618). Additional support for X-ray Scattering and TEM studies was provided by the US DOE, Office of Science, Office of Basic Energy Sciences, under Award No. DE-SC0020884. H.D. acknowledges Swiss National Science Foundation (SNSF) for granting him the Early Post-doc.Mobility (P2ELP2_191667) and Postdoc.Mobility (P500PN_214226) fellowships.

## Author contributions

H.D., E.W.M., and S.I.S. conceived the project. H.D. performed most of the experiments and analyzed the results. R.Q. and H.S. performed the X-ray scattering and negative-staining TEM characterizations. X.L. conducted the HDX-MS measurements. S.A.H.J., A.J.M., and H.D. developed the thermodynamic models for the micelle-to-polymer transformation. Y.W. did the high-throughput cross-correlation analysis of TIRF images. E.W.M. and S.I.S. supervised the research. H.D., E.W.M., and S.I.S. wrote the manuscript. All the authors contributed to the analysis of the results and provided input for the writing.

## Competing interests

The authors declare no competing interests.
