## [Transparent Peer Review file · Nature Communications]

Mapping In Situ the Assembly and Dynamics in Aqueous Supramolecular Polymers

Corresponding Author: Professor E.W. Meijer

Version 0:

Reviewer comments:

Reviewer #1

(Remarks to the Author)

The manuscript has undergone major revisions in response to earlier reviewer comments. The revised version effectively addresses previous concerns related to unclear justifications of key new insights, inadequate referencing, ambiguous or missed interpretations, and minor errors. The revised version emphasizes the significance of the combinatorial titration approach in revealing the mechanism of the assembly process—especially its ability to capture co-assembly behavior. This methodology has provided valuable insights such as revealing that the transition from random-coil micelles to β -sheet occurs without the involvement of other secondary structures. Compared to the earlier submission, the current version further demonstrates the method's capability to systematically monitor the influence of various components, including NaCl and PBS buffer, thus elucidating a deeper understanding of supramolecular polymerization compared to the other reported works. Given that such an in-depth analysis of the supramolecular co-assembly mechanism using combinatorial titration has not been previously reported, the revised manuscript could be suitable for publication in Nature Communications.

Reviewer #2

(Remarks to the Author)

I originally reviewed this manuscript for Nature Nanotechnology where I did not recommend publication, primarily due to literature precedent and the high requirements of that journal for novelty which I felt this paper did not meet. However, for Nature Communications, and with revisions made by the authors, I would recommend this paper as suitable for publication. Overall, many of the changes made by the authors have significantly improved the manuscript, however, there are a couple of issues I would like to mention for the authors' further consideration.

Most significantly is the nature of the deprotonation event which drives the assembly process. In their response letter, the authors refer to this as 'catastrophic', in the language of the paper, it is described as a highly cooperative deprotonation event. The argument is then that this highly cooperative deprotonation drives assembly. However, when I look at the pH titration in Figure 3e, which provides insight into the molecular deprotonation, I don't see this looking like a highly cooperative process. I agree that the ASSEMBLY process (as evidenced by CD in Fig. 3c) looks cooperative but I just don't see it in the pH titration data, which seems quite broad and shallow. A large relative amount of base has to be added to drive assembly, and the pH changes are very variable between different PAs. I agree that there is likely a point of deprotonation that is reached where the system starts to assemble, and then it assembles into fibrils cooperatively. I suspect this will also change the subsequent pKa values. But I am uncertain about the evidence that this assembly is being driven by a cooperative deprotonation event. Given cooperative deprotonation is mentioned many times in the paper, the authors must clearly present the evidence if they want to stick with this explanation. This also goes back to why I found it strange to discuss the CD evidence first, because this leads the authors down the path of talking about cooperative deprotonation before they have even discussed the pH titration curves for the system, which look far more complex.

The authors stated in their response that monomers are always CD silent, and only on assembly can CD signals be observed. This is not the case - molecules can show their own CD signals, particularly if they have internal helical structure organising chromophoric units. This is particularly the case for some small-medium helix-forming peptides and also

molecules such as binpathalenes etc. However, I fully accept their assumption here, given the low CMCs of the system being studied, which would make any contributions to CD from individual molecules essentially irrelevant.

The reference list has been significantly improved by adding some of the other suggested variable pH studies of fibril assembly. This includes the paper using variable pH studies to explore multi-component systems (ref 43), although I would note that the way this example has been included somewhat obscures the fact it included variable pH combinatorial methods to gain detailed insight, by focussing instead on other aspects of the work.

The NMR studies are a significant and useful addition to the paper, and the authors should be congratulated on making the effort to perform these studies.

I think I am convinced by the authors work/arguments about blockiness of the assembly. Certainly the schematic figures help the reader understand what they mean. It is really tricky given some of the studies use labelled peptides, some unlabelled, and the evidence is variable from study to study, and nothing is absolutely clear-cut. Personally, I think this emphasises just how tricky it remains to fully understand the complexity of multi-component assembly processes, even with state-of-the-art methods.

Overall, therefore I am happy to recommend publication of the work in Nature Communications subject to the authors considering the points above.

Reviewer #3

(Remarks to the Author)

The authors have responded sincerely to the more critical comments from Reviewers 1 and 2, and have substantially improved the manuscript. As a result, technical aspects that were previously unclear in the original version—partly due to space limitations—have now been clarified, making the manuscript a more comprehensive and complete research article. Since I was already supportive of its publication when it was submitted to Nature Nanotechnology, I also endorse the revised version for publication in Nature Communications.

Point-by-point answers to the comments of the reviewers

Our point-by-point responses are provided below (in blue). The corresponding revisions are highlighted in the annotated manuscript and are also summarized here.

Reviewer #1 (Remarks to the Author):

The manuscript has undergone major revisions in response to earlier reviewer comments. The revised version effectively addresses previous concerns related to unclear justifications of key new insights, inadequate referencing, ambiguous or missed interpretations, and minor errors. The revised version emphasizes the significance of the combinatorial titration approach in revealing the mechanism of the assembly process—especially its ability to capture co-assembly behavior. This methodology has provided valuable insights such as revealing that the transition from random-coil micelles to β -sheet occurs without the involvement of other secondary structures. Compared to the earlier submission, the current version further demonstrates the method's capability to systematically monitor the influence of various components, including NaCl and PBS buffer, thus elucidating a deeper understanding of supramolecular polymerization compared to the other reported works. Given that such an in-depth analysis of the supramolecular co-assembly mechanism using combinatorial titration has not been previously reported, the revised manuscript could be suitable for publication in *Nature Communications*.

We greatly appreciate the reviewer's positive and encouraging evaluation of our revised manuscript. We thank the reviewer for supporting the publication of our work in *Nature Communications*.

Reviewer #2 (Remarks to the Author):

I originally reviewed this manuscript for *Nature Nanotechnology* where I did not recommend publication, primarily due to literature precedent and the high requirements of that journal for novelty which I felt this paper did not meet. However, for *Nature Communications*, and with revisions made by the authors, I would recommend this paper as suitable for publication. Overall, many of the changes made by the authors have significantly improved the manuscript, however, there are a couple of issues I would like to mention for the authors' further consideration.

We thank the reviewer for the thoughtful comments and for recommending our revised manuscript for publication in *Nature Communications*. We are pleased that the revisions and additional experiments have addressed your previous concerns, and we appreciate your recognition of these efforts. Below, we provide our responses to the remaining points raised by you.

1. Most significantly is the nature of the deprotonation event which drives the assembly process. In their response letter, the authors refer to this as 'catastrophic', in the language of the paper, it is described as a highly cooperative deprotonation event. The argument is then that this highly cooperative deprotonation drives assembly. However, when I look at the pH titration in Figure 3e, which provides insight into the molecular deprotonation, I don't see this looking like a highly cooperative process. I agree that the ASSEMBLY process (as evidenced by CD in Fig. 3c) looks cooperative but I just don't see it in the pH titration data, which seems quite broad and shallow. A large relative amount of base has to be added to drive assembly, and the pH changes are very variable between different PAs. I agree that there is likely a point of deprotonation that is reached where the system starts to assemble, and then it assembles into fibrils cooperatively. I suspect this will also change the subsequent pKa values. But I am uncertain about the evidence that this assembly is being driven by a cooperative deprotonation event. Given cooperative deprotonation is mentioned many times in the paper, the authors must clearly present the evidence if they want to stick with this explanation. This also goes back to why I found it strange to discuss the CD evidence first, because this leads the authors down the path of talking about cooperative deprotonation before they have even discussed the pH titration curves for the system, which look far more complex.

We thank the reviewer for raising this critical point. We would like to clarify the meaning of “cooperative deprotonation” as used in our manuscript.

In line with previous work on supramolecular assembly of block copolymers (Li *et al.*, *Nat. Commun.* 7, 13214 (2016)), we concluded the cooperative deprotonation based on fitting the assembly process using an allosteric binding model, as detailed in Supplementary Text 2. This fitting yields both a Hill coefficient (n_H) and an apparent binding constant (K'). All three of our PA systems exhibit similar Hill coefficients of approximately 10, while differing in their K' values. A Hill coefficient significantly greater than 1 indicates a cooperative deprotonation process of micelles during their supramolecular polymerization/assembly process.

However, this does not mean that all micelles undergo deprotonation synchronously before reaching a global threshold to trigger a cooperative assembly, as suggested by the reviewer. Instead, the cooperativity we describe reflects a heterogeneous or localized mechanism, where only a subset of micelles preferentially undergoes a multi-step, cooperative deprotonation process, leading to polymer formation, while the rest remain as fully charged micelles. This interpretation is consistent with the report from Li *et al.* and aligns well with our proposed binary assembly model, which is convincingly supported by CD, TEM, and NMR results. This consistency also indicates that cooperative deprotonation and cooperative assembly are intimately linked with each other in our system.

We agree with the reviewer that the pH titration profiles differ among the PAs. This is also reflected in the fitted K' values, whose decreasing trend is consistent with the increasing apparent pK_a values. Theoretically, these two parameters can be related by the equation $pK_a + pK' = 14$. However, at the low micromolar concentrations of PA used here, bulk pH reflects not only PA deprotonation but also water dissociation and the dissociation of trace carbonic acid, even though we have taken very serious measures to minimize the latter (see Supplementary Fig. 3). Therefore, we use titrated [NaOH] as a reasonable proxy for $[OH^-]$, as detailed in Supplementary Text 2, and place greater weight on structure-specific techniques, such as CD and NMR, over pH titration data when extracting the cooperativity in the system. We hope the reviewer can understand our practical concerns here.

Regarding the “broad” pH range over which assembly occurs, we note that this range can be influenced by the degree of cooperativity. For example, in Li *et al.*, the molecule system exhibiting a much higher Hill coefficient (~ 51) results in a very sharp assembly response within a narrow pH window. In contrast, the lower n_H (~ 10) in our PA system likely lead to a broader pH range during assembly, despite still be cooperative.

To avoid further confusion, we have toned down our statement about “highly cooperative deprotonation” and added further clarification in the revised manuscript as follows:

A Hill coefficient significantly greater than 1 indicates a cooperative deprotonation process of micelles³⁸. Such cooperativity is expected to result in extensive deprotonation of a subset of micelles to form polymers with a negligible amount of intermediate charged states, while the remaining micelles stay fully charged. This mechanism is expected to facilitate a binary assembly model rather than a multistep assembly process, consistent with our experimental observations described above.

2. The authors stated in their response that monomers are always CD silent, and only on assembly can CD signals be observed. This is not the case - molecules can show their own CD signals, particularly if they have internal helical structure organising chromophoric units. This is particularly the case for some small-medium helix-forming peptides and also molecules such as binaphthalenes etc. However, I fully accept their assumption here, given the low CMCs of the system being studied, which would make any contributions to CD from individual molecules essentially irrelevant.

We thank the reviewer for raising this important point and are pleased that you agree on the negligible contribution of free monomers, given the low CMCs of the studied PA systems.

Regarding the CD silence of monomers, we note that many small chiral molecules, including those containing chromophores, typically exhibit silent CD signals at their monomeric state. This has been consistently observed in temperature-induced supramolecular polymerization studies in organic

solvents by one of our groups, as shown in Fig. 1b of *Korevaar, P. A. et al. Nature* **481**, 492–496 (2012) and Fig. S1 of *Korevaar, P.A. et al. J. Am. Chem. Soc.* **134**, 13482–13491 (2012).

However, we also acknowledge that certain longer peptides, particularly those capable of forming internally folded structures, may display distinct CD signals even in their monomeric state. To address the reviewer's concern, we have removed our statement about monomer CD silence from the revised manuscript.

Main manuscript

Circular dichroism experiments on the PA molecule C₁₆V₃A₃K₃ (PA-1, Fig. 1b) reveal the presence of random coils with their characteristic peak at 195 nm at a total monomer concentration ([PA]) of 20 μM (Fig. 2a), in contrast to the CD spectra of β-sheet polymers typically found in PA systems¹⁷.

Caption of Supplementary Fig. 24

Free monomers are expected to exhibit a distinct molar ellipticity (θ) compared to micelles. Therefore, a detectable presence of free monomers in PA systems would be expected to decrease the absolute values of θ with the decreasing total monomer concentration ([PA]). However, the values of θ in all three PA systems are nearly constant when [PA] is varied between 20 μM and 100 μM, indicating a negligible presence of free monomers here.

3. The reference list has been significantly improved by adding some of the other suggested variable pH studies of fibril assembly. This includes the paper using variable pH studies to explore multi-component systems (ref 43), although I would note that the way this example has been included somewhat obscures the fact it included variable pH combinatorial methods to gain detailed insight, by focussing instead on other aspects of the work.

To address the reviewer's comment, we have now also included this reference in the section discussing previous pH titration studies.

For a variety of aqueous assemblies, pH titrations are employed to investigate the charged systems^{6,22–32}, since their assembly states can be conveniently controlled through deprotonation and protonation by adjusting the pH.

4. The NMR studies are a significant and useful addition to the paper, and the authors should be congratulated on making the effort to perform these studies.

We thank the reviewer for having made this valuable suggestion and for recognizing our efforts to perform the additional NMR measurements as part of the revision.

5. I think I am convinced by the authors work/arguments about blockiness of the assembly. Certainly the schematic figures help the reader understand what they mean. It is really tricky given some of the studies use labelled peptides, some unlabelled, and the evidence is variable from study to study, and nothing is absolutely clear-cut. Personally, I think this emphasises just how tricky it remains to fully understand the complexity of multi-component assembly processes, even with state-of-the-art methods.

We are pleased that the reviewer finds our interpretation of the blocky nature of the assemblies convincing. We fully agree that completely elucidating the complexity of multi-component assembly processes remains a significant challenge, even when employing a combination of state-of-the-art characterization techniques. We are happy to have contributed a small step forward in this direction and believe that continued efforts by the research community will further advance understanding in this important area.

Overall, therefore I am happy to recommend publication of the work in Nature Communications subject to the authors considering the points above.

We thank the reviewer again for the positive feedback on our revision and for the support of our manuscript's publication *Nature Communications*.

Reviewer #3 (Remarks to the Author):

The authors have responded sincerely to the more critical comments from Reviewers 1 and 2, and have substantially improved the manuscript. As a result, technical aspects that were previously unclear in the original version—partly due to space limitations—have now been clarified, making the manuscript a more comprehensive and complete research article. Since I was already supportive of its publication when it was submitted to Nature Nanotechnology, I also endorse the revised version for publication in Nature Communications.

We sincerely thank the reviewer for the continued support and positive feedback. We appreciate the reviewer's endorsement of our work for publication in *Nature Communications*.